# Need for Orthodontic Treatment in Pupils Aged between 12 and 15 in the Valencian Region (Spain)

**DOI:** 10.3390/ijerph181910162

**Published:** 2021-09-27

**Authors:** José Enrique Iranzo-Cortés, José María Montiel-Company, Carlos Bellot-Arcís, Teresa Almerich-Torres, José Manuel Almerich-Silla

**Affiliations:** Stomatology Department, University of Valencia, 46010 Valencia, Spain; j.enrique.iranzo@uv.es (J.E.I.-C.); jose.maria.montiel@uv.es (J.M.M.-C.); carlos.bellot@uv.es (C.B.-A.); jose.m.almerich@uv.es (J.M.A.-S.)

**Keywords:** oral health, orthodontic treatment need, IOTN, DAI

## Abstract

The World Health Organization recommends carrying out periodic epidemiological studies in order to provide a basis for the evaluation of the state of health of the population at any given time; in doing so, action strategies can be established for the treatment of different pathologies. The objective of this study is to evaluate the need for orthodontic treatment in adolescents at school aged between 12 and 15 in the Spanish autonomous region known as *Comunidad Valenciana* (hereafter: Valencian Region). A cross-sectional study was carried out on a sample of 539 12-year-old schoolchildren and 460 15-year-olds, respectively, selected by cluster sampling and representative of the school-aged population of the Valencian Region, using the IOTN-DHC, IOTN-AC, and DAI indices. The need for specific orthodontic treatment according to the IOTN-DHC was 12.6% at 12 years and 7% at 15. For the IOTN-AC and DAI indices, the treatment needs were 4.3% and 0.9% at 12 years and 30.1% and 20.9% at the age of 15. These results were similar to those obtained in the previous study carried out on the same target population. There was no significant association between the need for treatment and gender or social class. We conclude that the need for orthodontic treatment presents values similar to those obtained in 2010.

## 1. Introduction

It is difficult to define malocclusion as a single concept, since it is highly variable and depends on the individual to be examined, the culture and country to which that an individual belongs to, or indeed the prevailing fashion at the time. Furthermore, it is not quantified in the same way by the patient as by the professional who assesses it. Malocclusion should not be considered a disease, but an anomaly in dental development that leads to incorrect alignment and an abnormal fit between dental arches. For this reason, determining at what point this anomaly should be considered a pathology requiring treatment is a pending problem for the specialty, making it very difficult to strike a consensus [1].

Orthodontic treatment is primarily based on an individual diagnosis by the orthodontist. This makes the epidemiological application of the complete diagnosis difficult, making it essential to use indices that are reliable, valid, and easily applicable. Many indices of treatment have been developed over the years with the aim of classifying the distinct levels of malocclusion and thereby establishing treatment priorities in populations in specific healthcare systems that may include orthodontic treatment; the said indices can assist the specialist in planning treatment options, but among orthodontists, solid unanimity is lacking as to which treatment or set of treatments is the most appropriate [2].

Most of the published studies, as well as the research carried out previously on the population, which is the subject of the present study, have applied two main indices: the Dental Aesthetic Index (DAI) and the Index of Orthodontic Treatment Need (IOTN) [1]. These indices are the ones that have shown a greater consensus on the individual characteristics and occlusal features that must be evaluated to objectively establish the need for treatment [1].

The periodic carrying out of epidemiological studies that analyze the presence of different diseases and the need for treatment in a given population is a tool that enables the planning of strategies for their treatment [3]. In the case of oral health, the WHO proposes, in its latest guide for conducting epidemiological studies, recommendations for the exploration of caries and periodontal areas, but no index is suggested to assess the need for orthodontic treatment, since the Dental Aesthetic Index (DAI) that it recommended in the fourth edition [4] is no longer included in the last 2013 [3].

According to the World Health Organization, conducting periodic oral health surveys is justified by the need to know the health status and treatment needs of a population and monitor changes in morbidity levels, as well as to gauge trends in the different diseases that can be found in the oral environment [5]. These are cross-sectional observational studies that seek to provide an overall appraisal of the problem in order to plan preventive and therapeutic decisions; furthermore, treatment needs need to be well identified based on the results of the said studies, to develop health policies. In addition, such studies can monitor trends in oral health status and evaluate existing programs.

Following from this, the objective of this study is to assess the need for orthodontic treatment, as well as the differences that may exist depending on the gender or socio-economic status of the schoolchildren studied.

## 2. Materials and Methods

### 2.1. Design of the Study

To meet the stated objectives, a descriptive cross-sectional study was designed in which the key features and variables that define malocclusion were analyzed by direct exploration according to the IOTN and DAI indices.

### 2.2. Sample Size and Selection Criteria

The target population for the present study is the 12- and 15-year-old school population of the Valencian Community. The sample size and the selection of the sample was carried out in such as to be representative of the total number of schoolchildren in these age groups and, thereby, enabling reliable conclusions to be drawn for this population group. To determine the sample size we have based ourselves on the prevalence of the need for orthodontic treatment obtained in the last epidemiological study carried out in the Valencian Region, carried out in 2010 [6]. In this study, the prevalence detected of the need for treatment of 21.7% was found at 12 years and 14.1% at 15 years, respectively; a 95% confidence level and a precision level of 4% were established, thereby requiring a necessary minimum sample size of 398 12-year-olds and 289 15-year-olds, respectively.

The sample was selected by performing a cluster sampling. This selection is based on a probability sampling that is set up in and straddles two stages. In the first stage, from all schools with first and fourth grade classrooms of the E.S.O. (Spain’s standard mandatory secondary-level schooling). Fifty-nine schools in the Valencian Region (made up of three provinces) were randomly selected (5 in the province of Castellón, 32 in the province of Valencia, and 22 in the province of Alicante), acting as the primary sampling unit. After receiving the permits from the schools and the parents or guardians of the minors, the children in the classrooms of both of the specified grades of each school underwent a dental examination, with each child acting as a secondary sampling unit. The response rate was 82.3%.

All 12 and 15-year-old schoolchildren present in the first- and fourth-grade secondary-school classrooms respectively of each school location where dental examinations took place were included. The schoolchildren excluded were those who did not present the pertinent authorization from their parents or legal guardians. For the intraoral examination of the occlusal features, those who were orthodontic appliance wearers at the time of the examination were also excluded.

### 2.3. Calibration Prior to the Study

Within the 2018 Study of the State of Oral Health in the Valencian Community [7], three graduates in dentistry were selected to carry out the dental examinations. These examiners were trained so that their clinical evaluations were consistent; they were given an “Dental Examiner’s Notebook” with all the guidelines and rules they had to follow, as well as the diagnostic criteria for the diagnosis of caries and the calculation of the DAI and IOTN indices. Once they had studied it, theoretical sessions were held to clarify doubts and the criteria were unified. In these sessions, a calibration was also carried out on orthodontic models that showed different levels of malocclusion and need for treatment. The inter-examiner reliability with respect to an expert orthodontic examiner was calculated for the determination of the DAI with the intraclass correlation coefficient, as it is a continuous variable; for the IOTN-DHC weighted Kappa statistics were used, as it is a categorical variable, obtaining in all cases a result with a value greater than 0.9.

### 2.4. Authorizations

First, a letter was sent to the schools to request authorization and establish the date for each child’s dental examination. The envelope included an informed consent with all the necessary information about the purpose of the examination and the procedure of the examination that would be carried out on the minor should authorization be granted. This consent form was sent to the parents or legal guardians of the schoolchildren and had to be returned, duly completed and signed, before proceeding with the examination.

This study has been approved by the Ethics Committee for research in humans of the University of Valencia, registered with reference number H1510648717945, respecting the ethical principles of the Declaration of Helsinki and complying with current regulations regarding data protection.

### 2.5. Material Used

The material used for the examination was that recommended by the WHO in its guide for conducting epidemiological surveys [5]. This included a Number 5 flat intraoral mirror and a PCP 11.5B periodontal probe. For each examination, a pair of disposable nitrile gloves was used, in anticipation that a minor could be allergic to latex, and a mask. A bluish-white spectrum lamp was used for illumination. At the end of the examination day, the material was disinfected and bagged for later sterilization in autoclave mode at the Dental Clinic of the University of Valencia. The findings were recorded on the examination sheets designed for the study, noting the measurements of the malocclusion features necessary to determine the need for orthodontic treatment indices.

### 2.6. Data Collection

The examinations were carried out in a designated classroom of the schools themselves. The examiners had been trained to collect data according to WHO recommendations, seeking the best possible lighting, location, and ergonomic conditions. The examination was carried out face-to-face with the child, both sitting respectively in a chair and with the minor extending their neck appropriately. The scorer filled out the examination sheet as the examiner proceeded with the examination.

To collect data on malocclusion features, a specific evaluation form for dentofacial anomalies evaluation form was used, designed for the 2010 Study of the Oral Health Status in the Valencian Community [6]. This form was designed considering those malocclusion features necessary for the calculation of the IOTN and DAI indices. All scans were carried out between April and October 2018.

### 2.7. Data Processing and Statistical Analysis

The collected data were included in a Microsoft^®^ Excel^®^ spreadsheet (Microsoft^®^ Corp., Redmond, WA, USA), following the same format as the dental examination sheets and the questionnaire, to facilitate the process. This database was exported to the IBM SPSS ^®^ v.24 (IBM, Armonk, NY, USA) program for statistical analysis.

First, a descriptive statistic was performed for each of the variables, analyzing the means for the quantitative variables and the proportions for the qualitative ones, with their respective 95% confidence intervals. To analyze statistically significant differences (*p*-value < 0.05), the Chi-square test was used in the case of proportions, and the Student’s *t*-test to determine the differences between means. In addition, the linear trend test was applied to assess the linearity between the categories.

To determine the socio-economic status, the occupations of the parents were recorded to infer the socio-economic status following the classification proposed by Domingo and Marcos [8]; following from this the socio-economic status of the child was that taken as the highest obtained by either of the two. This classification categorizes the professions as follows:I. Professionals, senior managers, and senior technicians;II. Other managers, middle-level technicians, and commerce;III. Intermediate-level and administrative positions;IV (a). Skilled manual workers;IV (b). Semi-skilled manual workers;V. Unskilled workers;“Not classifiable” was the category denoting those without a declared or a wrongly declared profession, wrongly classified, or belonging to the armed forces.

These categories were regrouped following the method proposed by the British Registrar General [9], thus considering groups I and II as having a high socio-economic status, category III representing an intermediate socio-economic status, while groups IV and V were categorized as having a low socio-economic status.

During the analysis, the variables DAI, IOTN-DHC and IOTN-AC were designated as: “Need for treatment: Yes”/“Need for treatment: No.” This dichotomization was carried out considering the instructions of the indices, which define treatment as necessary when the DAI index is greater than 30, the IOTN-DHC presents a value of 4 or 5 and the IOTN-AC can be identified as appearing to fall between photos 8 and 10.

## 3. Results

### 3.1. Population Examined

The total sample examined for the study on the need for orthodontics in schoolchildren in the Valencian Community was a total of 1,166 schoolchildren; there were 632 12-year-olds (54.2% of the total examined), whilst there were 534 15-year-olds (45.8%). Among the 12-year-old schoolchildren, 304 were boys (48.1%) and 328 girls (51.9%), while at the age of 15 there were 250 boys and 284 girls (46.8% and 53.2%, respectively).

### 3.2. Wearers of Orthodontic Appliances

In the 12-year-old group, 93 schoolchildren (14.8% of the total sample, 95% CI = 12.2–17.7%) were orthodontic patients at the time of the examination, 492 of the subjects examined (77.8%, 95% CI = 77.4–80.9%) had never worn orthodontics, and 47 (7.4%, 95% CI = 5.6–9.7%) had worn an orthodontic appliance it in the past. At 15 years, 74 children were having orthodontic treatment at the time of exploration (13.8% 95% CI = 11.2–17.0%), 323 had never received orthodontic treatment (60.5% 95% CI = 60.45%), and 137 had already finished their treatment (25.7% 95% CI = 22.1–29.5%).

To conduct this study, the schoolchildren who were wearing orthodontics at the time of the examination were excluded (n = 167), leaving a valid sample for the study of 999 pupils, 539 at 12-year-olds (265 boys, 274 girls), and 460 15-year-olds (222 boys and 238 girls) (Figure 1).

### 3.3. Socio-Economic Status

The socio-economic status was determined according to the profession of the parents, and the highest socio-economic status held by either of the two parents was the one taken as the determinant of the child’s socio-economic status. At 12 years, 24% (n = 152, 95% CI = 20.9–27.5%) of the schoolchildren belonged to the group with the lowest socio-economic status. For the intermediate-level of socio-economic status the percentage was 37.5% (n = 237, 95% CI = 33.8–41.3%). The group with the highest socio-economic status accounted for 38.4% of the sample (n = 243, 95% CI = 34.7–42.3%). On the other hand, at 15 years the percentages were 22.7% (95% CI = 19.3–26.4%), 40.1% (95% CI = 36.0–44.3%), and 37.3% (95% CI = 33.3–41.4%) for the low, middle, and high levels of socio-economic status, respectively (n = 121, 214, and 199).

### 3.4. Traits of Malocclusion

Table 1 shows a summary of the malocclusion features registered at both 12 and 15 years of age.

### 3.5. Treatment Need: The IOTN-DHC Indices (IOTN Dental Health Component)

Table 2 shows the distribution of the different degrees of IOTN in the examined sample at 12 and 15 years of age.

As indicated in the table, the vast majority of the examined population does not require orthodontic treatment according to the IOTN-DHC (87.4% of schoolchildren at 12 years old and 93% at 15 years old are between grades 1 and 3). Of the 12.6% who require orthodontic treatment at 12 years of age, only 2% have grade 5 of the IOTN-DHC, while, at the age of 15, of the 7% who require treatment, only 1.1% of children examined presented a very serious malocclusion.

### 3.6. Treatment Need: IOTN-AC Indices (IOTN Aesthetic Component)

The aesthetic component of the IOTN is determined from photographs, with which the patient identifies, to detect the perception and assessment that the patient has of their own dental aesthetics. The results corresponding to the 10 levels, for both age groups, can be seen in Figure 2.

These codes can be grouped into “no need for treatment” for photos 1 to 4, “moderate need for treatment” for photos 5 to 7, and “need for mandatory treatment” when the perception of their aesthetics is placed between photos 8 and 10. According to this classification, most children do not need treatment (88.5% (95% CI = 85.5–90.9%) at 12 years and 98% (95% CI = 96.3–99.0%) at 15), 7.2% (95% CI = 5.3–9.7%) of children at 12 years and 1.1% (95% CI = 0.5–2.5%) at 15 presents moderate need, and 4.3% (95% CI = 2.9–6.3%) and 0.9% (95% CI = 0.3–2.2%) present mandatory need for treatment at 12 and 15 years, respectively.

### 3.7. Treatment Need: DAI Index

The mean DAI index of 12-year-old schoolchildren was 27.87, with a 95% CI between 27.03 and 28.70, while at 15 years it was 25.01, with an interval ranging between 24.18 and 25.85. The distribution of the different degrees of need for treatment, after categorizing the DAI index for both age groups, can be seen in Table 3.

At 12 years of age, the mean DAI index for children was 27.43, with a 95% confidence interval between 26.33 and 28.53. For girls in this age group, the mean was 28.29, with a confidence interval between 27.04 and 29.54. On the other hand, at 15 years the means were 25.85 (95% CI = 24.59–27.12) and 24.23 (95% CI = 23.13–25.33) for boys and girls, respectively. When performing the Student’s *t*-test to contrast the means, it is observed that there are no statistically significant gender differences neither at 12 years (*p*-value = 0.312) nor at 15 years (*p*-value = 0.056).

### 3.8. Differences between Gender and Socio-Economic Status

Table 4 shows the comparison between the need for treatment according to the different indices based on the age group and the gender of the schoolchildren examined. In this comparison, the Chi-square did not show statistically significant differences related to gender in either of the two age groups. When comparing the need for treatment according to socio-economic status, for both age groups, we observe that there are no statistically significant differences between groups in most cases, only a slight significant difference being found in the IOTH-AC at 15 years.

## 4. Discussion

Any epidemiological study that is carried out must have, as one of its main requirements for the results obtained to be reliable, a fully representative sample of the target population on which conclusions are to be drawn from the data obtained. To do this, the sample must be selected randomly from a group of individuals, representing the entire target population of the study. This type of sample selection is called a controlled bias sample. In the case of the present study, given that the individuals of the target population are organized into small groups (classrooms in schools), the most appropriate type of sampling is cluster sampling.

The sample obtained for this study was obtained random selection of the Secondary Education schools of the Valencian Community, this method being the one used in other previous studies such as those of Onyeaso et al., 2005, Bravo-Pérez et al., de Paula et al., 2011, or Almerich-Silla et al. of 2004, justifying its representativeness [6,10,11,12,13]. However, the studies by Kok et al., 2004, Ngom et al., 2007, or Sardenberg et al. of 2011 [14,15,16] use another methodology, since they analyze a convenience sample depending on the place of study, such as an institute, a health center, or a specific dental school.

Most of the studies carried out to evaluate the need for treatment exclude in their methodology individuals with orthodontics or who have been carriers of some type of orthodontic appliance, as it is impossible to know the need for treatment of these patients before starting the procedure [17,18,19,20]. Our study has only eliminated schoolchildren who currently wore orthodontics, not excluding those who wore it in the past. After this exclusion (167 schoolchildren in total), the sample was 539 12-year-old children and 460 of 15. Since only patients who currently wore orthodontics were excluded, the results and conclusions obtained are more reliable for the group of 12 years, since at 15 there were patients who had already finished orthodontic treatment, so the values of need for treatment in these patients are slightly biased. This limitation is similar to that which occurs in the estimates of caries indices in oral health surveys, since when a filling is detected, it is assumed that treatment took place due to a previously existing caries, assuming the restorative procedure to be justified, when in fact there is no way of knowing for sure if this was the case or not.

Only one study, published by Brook and Shaw in 1989 [21], has been found in which the assessment of the need for orthodontic treatment of study subjects who had received orthodontic treatment in the past was carried out by analyzing the models prior to the start of treatment, thus avoiding the aforementioned bias. This methodology could be used because the study was carried out in a country where the majority of orthodontic treatments are carried out by specialists within the framework of public health, and there is a defined protocol regarding record-taking. This would be impossible to develop in our country, since there is a wide variety of specialized professionals who perform orthodontic treatments in the private sector.

No changes have been observed in the percentage of children with orthodontics either at 12 years of age (12.9% of the population explored in the 2010 survey), or at 15 (15% in 2010).

According to the DAI index, the percentage of 12-year-old children requiring treatment is 30.1%, with a mean index of 27.85. These results are higher than those obtained in the study by Almerich et al. [6], carried out on the same target population. If we look at other published works, we find studies with lower values of the DAI index [22,23,24,25,26], similar values [27] or higher values [6,28,29,30,31].

On the other hand, at 15 years of age, the percentage of children in need of treatment is 20.9%, with a mean DAI index of 25.01. Compared with other studies, these results are lower [6,18,26,28,32,33,34], similar [6,24,27,35,36], or higher than the publications found [6,25,31].

In the present study, the percentage of 12-year-old schoolchildren requiring orthodontic treatment according to the IOTN-DHC is 12.6% and 4.3% when the IOTN-AC is used. On the other hand, at 15 years of age, the percentage of schoolchildren who required orthodontic treatment according to the IOTN-DHC was 7%, with the need determined by the IOTN-AC being 0.9%. These results are lower than those obtained in most of the studies analyzed in the literature review, except that of Otuyemi et al., Cooper et al., Unçüncü et al., Alkhatib et al., Mandall et al., Souames et al. al., Hedayati et al., Svedström-Oritsto et al., Emerich et al., Chaitra et al., Jamilina et al., or Tsiouli et al., whose results were similar or lower than those obtained in our results [17,22,37,38,39,40,41,42,43,44,45,46].

In the study by Woon et al. it was suggested that ethnic differences could affect the degree of malocclusion present in different populations [47]. This statement is not conclusive, since multiple subsequent works, such as those by Burden et al. from 2001 or Ngom et al. of 2007, do not find an association between the need for treatment and the ethnicity of the patients [15,48] or, if they find differences between ethnic groups, these are not significant, as observed in the study by Josefsson et al. [49].

In this regard, Ngom et al. stated in their 2005 study that not only the degree of malocclusion is linked to ethnicity, but also to the patient’s perception of their own malocclusion [50]. In contrast, Otuyemi et al. in 1999 or Onyeaso et al. suggest that the concepts of beauty are not linked to racial or cultural differences, with the exception of a specific feature such as the interincisive diastema in the Nigerian population [11,22].

In addition, another possible cause of discrepancy between results obtained between different studies could be the cut-off point used to determine the need, since we can find studies with high values of need for treatment, such as those obtained by Ghijselings et al. [51], finding a need for treatment of 80.3% for the IOTN-DHC and 38.3% according to the IOTN-AC. These authors determined that the need for treatment was found when the IOTN-DHC codes were equal to or greater than 3 and the IOTN-AC codes were equal to or greater than 5, in contrast to the methodology used in most studies, which follow the recommendations of Burden et al. [48], marking the need for treatment for IOTN-DHC when the code is 4 or 5 and for IOTN-AC when it is equal to 8 or higher.

It can be argued that the demand for orthodontic treatment is directly related to income or socio-economic status, this statement was accepted by Proffit et al. [52], who concluded that those families with higher incomes can more easily afford orthodontic treatment expenses. In addition, they stated that the absence of malocclusions improves facial appearance, associating this with a better social position. These authors also suggested that, in the lower income groups, approximately 5% of adults receive orthodontic treatment, this percentage reaching 10–15% in the middle-income group. They considered that the effect of financial difficulties on the demand for orthodontic treatment was clearly seen in response to private insurance cover in that demand increased considerably when insurance was available to pay for part of the treatment [19].

Bravo et al. [53] considered that the perceived need was determined according to social and cultural conditions. They found that families with a high socio-economic level more frequently appreciated dental crowding and were more willing to solve them, especially in those countries where orthodontic treatment is not included in health benefits and must be paid for privately.

In our results we found that, when analyzing the need for treatment, no differences were found based on socio-economic status. These results do not correspond to those obtained in other studies [2,20,54]; however, although the statistical tests have not shown significant differences, a certain trend can be observed, with the need for treatment being greater in the lower socio-economic levels both at 12 and 15 years, compared to the highest socio-economic level.

The studies by Gazit-Rappaport et al. and Hassan et al. (2010), and Puertes-Fernández et al. (2011), analyzed the need for treatment based on gender and concluded that there are no statistically significant differences between the two groups [25,55,56]. In the case of the study by Jossefson et al. the extreme of not looking for differences in the need for treatment between boys and girls was reached due to the low sample size, not allowing therefore sufficient robustness in the statistical study, and justifying it via the evidence of the aforementioned publications [49]. However, we did find other studies that showed statistically significant differences between boys and girls, although they could not justify these results [20,24,30,42,57].

In the present study, no differences were found between the sexes in any age group, nor for the IOTN-DHC indices (Chi square test; *p* = 0.36 and *p* = 0.35 at 12 and 15 years, respectively) or IOTN- AC (*p* = 0.77 and *p* = 0.35 at 12 and 15 years) or for DAI (*p* = 0.29 and *p* = 0.39 at 12 and 15 years).

### Limitations and Strengths of the Study

Both in the studies carried out to assess the need for treatment in Spain and in other international ones, the DAI and IOTN indices have been the most widely used [1]. These indices have also been used in other studies previously carried out in the Valencian Region [19,20]. The choice to use the DAI and the IOTN as indices to evaluate the need for orthodontic treatment in the present study has been justified as they both meet the fundamental requirements of epidemiological indices: validity, proven reliability, simplicity, speed of use, the possibility of use by non-specialized personnel, and objectivity [1,16,20,58,59,60]. All the examiners obtained excellent results in the intraclass correlation coefficient for the determination of DAI and used weighted kappa for the determination of the IOTN with respect to the Gold Standard.

We have carried out a cross-sectional study in which 12- and 15-year-old children have been examined, carrying out the examinations in the same school, which may limit record-taking; however, the WHO recommendations have been followed [3] to minimize this limitation. Furthermore, due to the very characteristics of the sample selection, there a selection bias may be present; this entails—due to chance—leaving a part of the population outside the studied sample that could present high levels of the disease. In any case, the sample size from which the statistical analysis was performed is large enough to assume that the selected sample is representative of the total population and this is one of the strengths of the study. In addition, a high response rate was obtained from the schools involved with 82.3% of the schoolchildren accepting to participate in the study.

The determination of socio-economic status using the classification by Domingo and Marcos [8] should be reviewed, since it has been shown to be one of the weaknesses of our study. Socio-economic status is determined by income and educational or schooling level. At the time of creation of the classification, both were united in most cases, but nowadays this is not always the case, following from this, socio-economic status should not be determined solely by occupation, since this can indicate a high level of formal education but not necessarily accompanied by a high income that would otherwise identify the family as one with high socio-economic status.

## 5. Conclusions

After conducting the present study, we can conclude that, when the IOTN-DHC index is used, the need for treatment at 12 years was 12.6%, while at 15 years was 7%. However, when the need for treatment was determined by the IOTN-AC, the need was 4.3% at 12 years and 0.9% at the age of 15. When using the DAI index, the need for orthodontic treatment found at 12 years was 30.1%, while at 15 it was 20.9%. No statistically significant differences were found between the need for orthodontic treatment based on gender or socio-economic status, although in the latter case it was found that the need increased in schoolchildren with lower socio-economic backgrounds.

## Figures and Tables

**Figure 1 ijerph-18-10162-f001:**
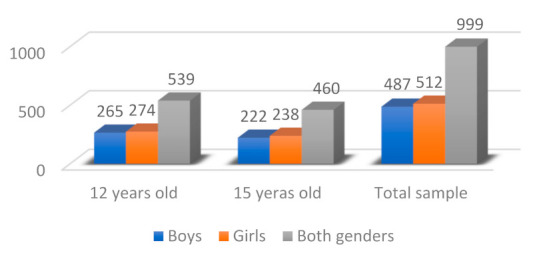
Valid sample distribution by age and gender.

**Figure 2 ijerph-18-10162-f002:**
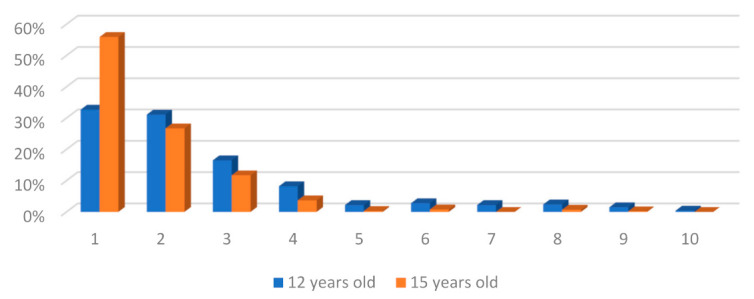
Distribution of the 10 grades of treatment need according to the aesthetic component of the IOTN (IOTN-AC).

**Table 1 ijerph-18-10162-t001:** Summary of the values found for the traits of malocclusion (CI95%).

	12 Years Old	15 Years Old
Overjet/reverse overjet	3.34 (3.16–3.52)	2.61 (2.45–2.76)
Overbite/openbite	3.52 (3.33–3.71)	2.88 (2.72–3.05)
Midline shift	0.65 (0.56–0.74)	0.59 (0.50–0.68)
Incisal crowding 1 segment	28.8% (25.1–32.7%)	22.8% (19.2–26.9%)
Incisal crowding 2 segments	30.8% (27.0–34.8%)	22.0% (18.4–26.0%)
Incisal spacing 1 segment	17.5% (14.5–20.9%)	11.6% (8.9–14.8%)
Incisal spacing 2 segments	6.1% (4.4–8.5%)	3.0% (1.8–5.0%)
Inter-incisal diastema	0.32 (0.25–0.39)	0.18 (0.13–0.23)
Highest maxillary irregularity	1.61 (1.46–1.77)	1.20 (1.05–1.36)
Highest mandibular irregularity	1.41 (1.30–1.53)	1.12 (1.01–1.25)
Unilateral posterior cross-bite	10.4% (8.1–13.3%)	8.9% (6.6–11.9%)
Bilateral posterior cross-bite	2.2% (1.3–3.9%)	2.4% (1.3–4.2%)
Molar class II right side	23.2% (19.8–26.9%)	20.9% (17.4–24.8%)
Molar class II left side	25.5% (21.9–29.3%)	4.6% (3.2–6.8%)
Molar class III right side	5% (3.5–7.2%)	4.8% (3.2–7.1%)
Molar class III left side	23.3% (19.6–27.3%)	5.4% (3.7–7.9%)
Canine class II right side	23.2% (19.8–26.9%)	18.3% (15.0–22.0%)
Canine class II left side	26.2% (22.2–30.7%)	19.8% (16.4–23.7%)
Canine class III right side	5.2% (3.6–7.4%)	5.2% (3.5–7.6%)
Canine class III left side	4.7% (3.2–6.8%)	6.3% (4.4–8.9%)

**Table 2 ijerph-18-10162-t002:** Sample distribution in the different malocclusion grades, according to IOTH-DHC, at 12 and 15 years old (CI 95%).

		Age Group	
		12	15
Grade of malocclusion	Grade 1. No treatment need	20.2%	37%
		(17.0–23.8%)	(32.7–41.5%)
	Grade 2. Little treatment need	43.6%	38.4%
		(39.5–47.8%)	(34.1–43.0%)
	Grade 3. Borderline treatment need	23.6%	17.6%
		(20.2–27.3%)	(14.4–21.4%)
	Grade 4. Treatment required	10.6%	5.9%
		(8.3–13.5%)	(4.1–8.4%)
	Grade 5. Treatment required	2.0%	1.1%
		(1.1–3.6%)	(0.4–2.5%)

**Table 3 ijerph-18-10162-t003:** Distribution of the orthodontic treatment need grades according DAI, at 12 and 15 years old (CI95%).

		12 Years Old	15 Years Old
DAI grades	Grade 1: No treatment need/slight treatment need	52.5%	66.7%
		(48.3–56.7%)	(62.3–70.9%)
	Grade 2: Elective treatment	17.4%	12.4%
		(14.5–20.9%)	(9.7–15.7%)
	Grade 3: Highly desirable treatment	14.9%	10.5%
		(12.1–19.1%)	(7.9–13.6%)
	Grade 4: Mandatory treatment	15.2%	10.4%
		(12.4–18.5%)	(7.9–13.6%)

**Table 4 ijerph-18-10162-t004:** Orthodontic treatment need according gender or socio-economic status for the groups of 12 and 15 years old. *p*-value Chi^2^ test. * (*p* < 0.05).

			Gender			Socio-Economic Status			
			Boys	Girls	*p*-Value	Low	Intermediate	High	*p*-Value
Treatment need	12 years old	IOTN-DCH	14.0%	11.3%	0.36	18.0%	9.0%	12.6%	0.05
			(13.3–18.7%)	(8.1–15.6%)		(12.5–25.2%)	(5.9–13.7%)	(8.6–18.1%)	
		IOTN-AC	4.5%	4.0%	0.77	5.0%	2.9%	5.3%	0.43
			(2.6–7.7%)	(2.3–7.0%)		(2.5–10.0%)	(1.3–6.1%)	(2.9–9.4%)	
		DAI	27.9%	32.1%	0.29	34.5%	30.0%	26.8%	0.32
			(22.9–33.6%)	(26.9–37.9%)		(27.2–42.8%)	(24.2–36.5%)	(21.0–33.6%)	
	15 years old	IOTN-DCH	8.1%	5.9%	0.35	7.5%	8.1%	5.3%	0.57
			(5.2–12.5%)	(3.6–9.6%)		(3.9–14.2%)	(5.0–12.9%)	(2.8–9.8%)	
		IOTN-AC	0.5%	1.3%	0.35	2.8%	0.5%	0.0%	0.04 *
			(0.08–2.5%)	(0.4–3.6%)		(1.0–8.0%)	(0.01–3.0%)	(0.0–2.2%)	
		DAI	22.5%	19.3%	0.39	25.5%	18.4%	20.7%	0.36
			(17.5–28.5%)	(14.8–24.8%)		(18.1–34.5%)	(13.5–24.6%)	(15.3–27.4%)	

## Data Availability

Not applicable.

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
