# Peer review of "Need for Orthodontic Treatment in Pupils Aged between 12 and 15 in the Valencian Region (Spain)"

_ijerph, 2021, doi:10.3390/ijerph181910162_

Round 1

Reviewer 1 Report

The present study aimed to assess the need for orthodontic treatment, as well as the possible existence of differences in gender or socioeconomic status of the studied population. The data obtained are likely to help plan the local health care system to plan resources for timely and high-quality care for this patient population.

  1. I would like the authors to more clearly reflect on the period in which the study was conducted. It would be helpful to indicate the recommended frequency of such epidemiological studies for dental practitioners. This will strengthen the value of conducted work.
  2. In the materials section, the authors point to a periodontal probe and a mirror as the instruments used. Perhaps we are talking about a pair of sterile instruments for each patient. The protocol of infection control of the study raises doubts about its ethics, in particular, the examination of children in classrooms, and thus I would like to receive detailed comments from the authors on this issue.
  3. The approval for publication from the patients who participated in your study is mandatory. Please provide a blank form of consent for publication.
  4. As the paper reports the result of a randomized controlled trial, but does not include the randomized controlled trial registration number. Your manuscript should include the international clinical trials register and cite a reference to the registration in the Abstract and Methods section.

Author Response

Dear reviewer,

Thank you for your review.

Please, find the responses in the attachment.

Reviewer 2 Report

the study takes in itself a very significant sample and presents relevant results for the literature

The inclusion and exclusion criteria must be included in materials and methods

review some terms and repeated phrases in the text

Author Response

(The authors gave the same response as above.)

Reviewer 3 Report

Many methodological biases exist

(The Authors must see my remarks)

Author Response

Dear reviewer,

Thank you for your review. In the following lines, we will try to answer all your comments:

We have tried to make all the changes and corrections that you made to our article and to answer all your requests. You can find them in the document with your review.

We hope we have answered all your requests.

Round 2

Reviewer 3 Report

Methodological biases exist

Author Response

Dear reviewer,

Thank you for your review.

We have been reading the peer review and didn’t find any request for changes or suggestions for improvement.

If you think that we have to make any change, we will be grateful if you tell us what you think we can improve.

Yours sincerely